# Regional Spatiotemporal Patterns of Fire in the Eurasian Subarctic Based on Satellite Imagery

Yikang Zhou [1], Shunping Ji [1,*] and Timothy A. Warner [2]

1. School of Remote Sensing and Information Engineering, Wuhan University, 129 Luoyu Road, Wuhan 430079, China
2. Department of Geology and Geography, West Virginia University, Morgantown, WV 26506-6300, USA
* Correspondence: jishunping@whu.edu.cn

**Abstract:** The fire risks in the vast Eurasian Subarctic are increasing, raising concerns for both local and global climate systems. Although some studies have addressed this problem, their conclusions only draw from relatively lower resolution data, and the sub-regional analysis of fire patterns in this area is lacking. In this paper, using a huge amount of multi-temporal and multi-resolution remotely sensed data, derived products, and weather data between the period 2001 and 2021, we reveal several novel and recent findings concerning regional and overall fire patterns in the Eurasian Subarctic. First, we discovered that fire occurrence over the period 2001 and 2021 varied by sub-region within the Eurasian Subarctic, with perennial low fire incidence in the East European and West Siberian Plain, increasing fire incidence in the Central Siberian Plateau, and marked periodicity of fire in the East Siberian Highlands. Second, we reveal the larger scale of individual fires in the Eurasian Subarctic compared to the adjacent region to the south, with fires of longer duration (13 vs. 8 days), larger daily expansion area (7.5 vs. 3.0 $km^2/d$), and faster propagation (442 vs. 280 m/d). Third, the northern limit of fire has extended poleward approximately 1.5° during the study period. Fourth, the start dates of fire seasons in Eurasian Subarctic, dominated by the Central Siberian Plateau, has advanced at a rate of 1.4 days per year. We also analyzed the factors resulting in the regional patterns of fire incidence including weather, human activity, land cover, and landscape structure. Our findings not only increase the knowledge of regional fire patterns and trends in Eurasian Subarctic but also will benefit the design of special fire management policies.

**Keywords:** fire pattern; Eurasian Subarctic; wildfire; remote sensing; land cover

## 1. Introduction

High-latitude terrestrial ecosystems of the Subarctic region are experiencing increasing incidence of fire [1]. Climate change has led to increased lightning activity and drier vegetative and ground fuel conditions, which has, in turn, driven an increase in the numbers and size of arctic fires [1]. The increased incidence of Subarctic fires raises concerns for both the local region and the broader global climate system. The Subarctic preserves large quantities of organic-soil carbon, equivalent in magnitude to the carbon content of the earth's atmosphere [2–4]. Much of this organic-soil carbon resides across Eurasia and North America in Arctic tundra, tussock, and shrub ecosystems, as well as sparsely forested taiga ecosystems [5,6]. When organic-soil is burnt, stored carbon dioxide is released, forming a positive feedback loop. So-called latent fires, smoldering in carbon-rich peat below the Subarctic surface for months or years, are the largest fires on Earth in terms of fuel consumption [7]. Subarctic fires, along with climate change, have also led to changes in the ecosystems of the high northern latitudes. Tundra vegetation types and some fire-resistant ecosystems are becoming more likely to be burnt, such as tundra bogs, fens, and marshes [8]. In addition to carbon release and the positive feedbacks to climate change, increases in the frequency, magnitude, and severity of Subarctic tundra fires are contributing to increased

thermokarst development, permafrost degradation, and associated landscape change in Subarctic tundra regions [9].

Although there are major concerns regarding the impact of Subarctic fires on the global climate, ecosystems, and Subarctic communities, fire nevertheless provides important environmental benefits. Some species in the Subarctic ecosystem are fire-adapted, such as larch (*Larix Mill.*) and Scots pine (*Pinus sylvestris* L.), with periodic wildfires supporting ecosystem function, biodiversity, conservation, and reduced danger of catastrophic wildfires [10]. Therefore, designing sustainable fire management policies is a complex task, requiring detailed information regarding fire occurrence. In particular, documenting and analyzing the spatio-temporal characteristics of fire in the Subarctic can give important insight regarding how the region's fire regimes have changed and how they might evolve in the future. For example, using charcoal and pollen data from peat cores, Feurdean et al. [11] determined that the fire regimes of western Siberia were established approximately 1500 years ago and contrast markedly with those of the previous 3500 years. By monitoring and calculating the black carbon released by wildfires on Russian forest lands, Smirnov et al. [12] found that forest fires in Russia from 2008 to 2012 mainly occurred from June to August and that the main areas where forest fires occurred were Siberia and the Far East of Russia.

Remote sensing offers the potential to provide information on fire frequency and the extent of burning over wide areas, complementing field studies and regional statistical summaries. Nevertheless, there have been few studies that use remote sensing data to focus on the geographic variation of fire in the Eurasian Subarctic. Using MODIS data, York et al. [13] observed that sub-seasonal droughts over a period of weeks cause highly variable wildfire activity in the high northern latitudes. McCarty et al. [1], also drawing extensively on satellite data, linked biomass burning emissions with a changing fire regime for the Subarctic and found that wildfire had increased from 2010 to 2020, particularly above 60°N. In analyses based on burnt area, land surface temperature, and biomass burning emissions, both York et al. [13] and McCarty et al. [1] found that, though the extent of wildfires in Subarctic is highly variable from year to year, overall fire occurrence has increased in recent years. Talucci et al. [14] used 20 years of Landsat imagery to characterize fire regimes of the Siberian taiga and tundra. Their data showed that although 2003 was the largest fire year during the studied period, 2020 was exceptional for the occurrence of fire above the Arctic Circle. The research of Kirillina et al. [15], drawing on satellite observations and field data, shows that the average duration of fire seasons has increased by 13 days since 2013 in the Sakha Republic, Russia. In summary, previous studies have shown that fires are becoming more frequent in the whole or part of the Subarctic in terms of biomass burning and fire season duration. However, the Eurasian Subarctic covers a vast area, and the effects of changes in climate and land cover are unlikely to be uniform across the entire region. In particular, information on the regional differences in fire patterns, the role of land-cover type, and fine scale information on the propagation characteristics of individual fires is lacking.

In this work, we used remote sensing technology to explore the fire patterns of the Eurasian Subarctic between 60~85°N and 0~180°E. After investigating the availability of existing fire products, we selected the years 2001 to 2021 as the time period for the analysis. Considering the large extent of the Eurasian Subarctic and the large difference in physical geography and human activities from west to east, we divided the study area into three regions (Figure 1): the East European and West Siberian Plain (region A, 0~85°E, approximately 3,808,928 km$^2$), characterized by flat and swampy land; the Central Siberian Plateau (region B, 85~137°E, approximately 3,646,170 km$^2$), which has some of the largest unbroken forest tracts on Earth; and the East Siberian Highlands (region C, 137~180°E, approximately 2,216,819 km$^2$), which includes some of the largest mountain systems of Russia.

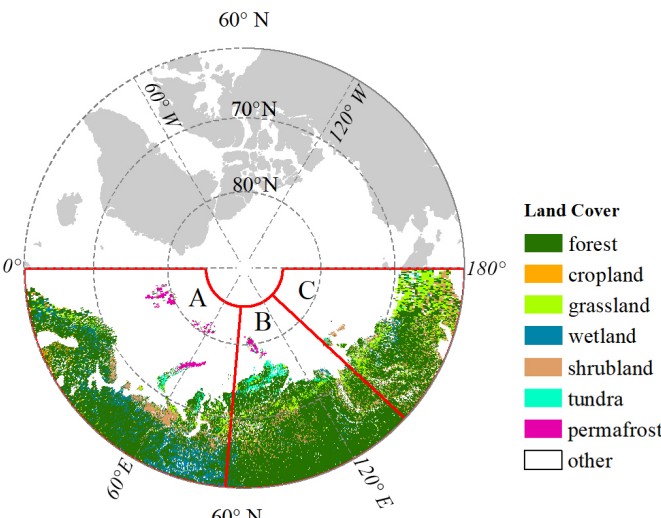

**Figure 1.** The Subarctic study area and the three regions. Data source: ESA Land Cover Data. A is the East European and West Siberian Plain; B is the Central Siberian Plateau; C is the East Siberian Highlands.

We documented fire occurrence within the region as a whole and also compared the three sub-regions. We explore the spatio-temporal distribution of fire frequency, burnt area, association of fire with different land cover types, duration of individual fires, propagation speed, and direction of fire fronts.

We grouped the remotely sensed data and derived products into three levels of resolution: coarse resolution, medium resolution and fine resolution data. The coarse and medium data draw upon published datasets, and the fine resolution data are based upon a custom fire front dataset, created by us. Through analysis at these three resolution levels, we provide a detailed description of the changing fire occurrence in the Eurasian Subarctic over the last two decades. Our study provides new discoveries of:

- The different patterns of fires in sub-regions of Eurasian Subarctic and the associated driving factors.
- Trends in the start-dates, duration of fire season, and the northern limit of fires.
- The association of fire behavior, including the speed of the propagation of fire fronts, with land-cover type.
- A comparison of fire behavior in the Eurasian Subarctic with the adjacent region to the south.

## 2. Materials and Methods

In this section, we introduce the remote sensing, biomass burning emission, fire weather, and land-cover time-series data sets (Table 1) used for the analysis and methods for processing the existing datasets and for creating new datasets.

### 2.1. Processing of Coarse Resolution Data

Coarse resolution data were used to characterize the spatio-temporal distribution of active fires and to estimate fire frequency. The primary data for this analysis were the 1 km MODIS Standard Fire product (MCD14ML) [16]. At the time of writing this article, MCD14ML was available from November 2000 (for Terra) and from July 2002 (for Aqua) to August 2021. MCD14ML includes confidence values for each pixel. For our analysis, we excluded fire pixels marked with low confidence, a total of 3.7% of the dataset.

The other coarse resolution dataset we used is the Global Fire Emissions Database Version 4.1 (GFEDv4) [17], which was derived from 500-m MODIS burned area maps and active fire data from Tropical Rainfall Measuring Mission (TRMM), visible and infrared scanner (VIRS), and along-track scanning radiometer (ATSR) family of sensors. GFEDv4

provides global estimates of monthly carbon emissions from biomass burning. We further calculated the annual biomass burning carbon emissions for the three regions from 2001 to 2021.

For the generation of maps of numbers of fire pixels per latitude band (Figure 2), it was necessary to account for the progressively reduced area with increasing latitude. The normalized number of fires pixels was calculated as $(n/A) \times S$, where $n$ denotes the number of fire pixels in a latitudinal band, A denotes the area of land covered by this band (km$^2$), and S is a scaling factor (1,000,000) to rescale the values to a more convenient range.

**Table 1.** Overview of the three scales (coarse resolution, CR, medium resolution, MR, and fine resolution, FR) of the fire analysis and the associated data used.

| Scale | Purpose | Period Covered | Revisit Time | Spatial Scale (Pixel Size) | Sensor | Product Used |
|---|---|---|---|---|---|---|
| CR | Fire incidence: daily and annual scale<br>Spatial distribution of fires; Fire season<br>Biomass burning emissions<br>Fire weather conditions | 2001–2021<br>2001–2021<br>1982–2018<br>1979–2020<br>1979–2021<br>1979–2021<br>2010–2021 | Daily<br>Monthly<br>Annual<br>Daily<br>Daily<br>Hourly<br>Hourly | 1 km<br>$0.25° \times 0.25°$<br>$0.05° \times 0.05°$<br>$1.0° \times 1.0°$<br>$0.25° \times 0.25°$<br>$0.1° \times 0.1°$<br>$0.5° \times 0.5°$ | MODIS<br>Various<br>AVHRR<br>-<br>-<br>-<br>- | MCD14ML<br>GFEDv4<br>FireCCILT11<br>GPCP v2.3<br>ERA5 reanalysis<br>ERA5-Land<br>WGLC |
| MR | Area burnt per year<br>Land cover | 2001–2020<br>2000–2019 | Annual [1]<br>Annual | 500 m<br>300 m | MODIS<br>Various | GlobFire/MCD64A1<br>ESA CCI |
| FR | Evolution of individual fires<br>Land cover | 2017–2021<br>2016–2020 | Average 5 days<br>Annual | 20 m<br>300 m | Sentinel-2<br>Various | Fire Front Dataset [2]<br>ESA CCI |

[1] The GlobFire database provides data at the daily scale; our processing summarizes the data at an annual scale.
[2] Custom dataset we created for this research.

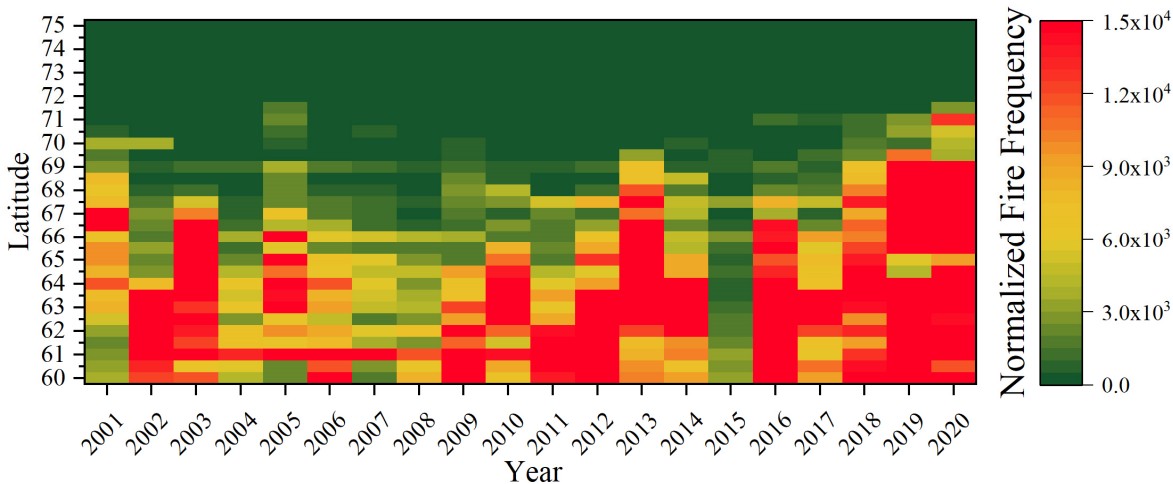

**Figure 2.** Spatial distribution of active fire pixels. Latitude vs. time (with number of fire pixels normalized to account for reduced area with increasing latitude).

### 2.2. Processing of Medium Resolution Data

The medium resolution data were used to provide information on the area burnt within different land cover categories, to supplement the time and location of fire information provided by the coarse resolution data, and to estimate carbon emissions from biomass burning. The burnt area by year for the period from 2001 to 2020 was obtained from the GlobFire database. The GlobFire database is generated using the MODIS MCD64A1 burn

area product, which provides an estimated burn date for each 500 m pixel [18]. GlobFire [19] tracks the day-to-day dynamics of individual fires by applying data mining to the MCD64A1 data. In addition, because each burned patch in the time series for an individual fire is designed to summarize the cumulative extent of the fire, simple summation of the burnt areas to estimate burnt area would double-count many pixels. We, therefore, conducted a union processing among all fire-stage layers in GlobFire so that pixels labeled as burnt were recorded only once for each individual fire.

In order to analyze the spatio-temporal distribution of fires on different land-cover types and their contribution to the total burnt area, we obtained the annual ESA Land Cover CCI Product [20]. This product has a nominal 300 m pixel size and is generated from six satellite-borne sensors, including AVHRR and MERIS. We resampled the 300 m data to 500 m, using the nearest neighbor method, and the 22 land cover types in the ESA product were aggregated to eight categories of land cover: forest, cropland, grassland, shrubland, wetland, tundra, permafrost, and other non-vegetation types (urban areas, bare areas and water bodies) [20]. For each year of burnt area data, we overlaid the land-cover data of the previous year, since the land-cover data of the year of the burn may record the post-burnt land cover instead of pre-burnt land cover. Thus, we used the annual land cover maps from 2000 to 2019 to compare to the land cover of areas indicated as burnt in the GlobFire data from 2001 to 2020. In this way we generated 20 pairs of maps from 2001 to 2020.

### 2.3. Definition of High Fire Incidence Year and Peak Fire Season

Our analysis mainly focuses on fire season trends in high fire incidence years. We used an empirically chosen threshold of 80% to differentiate high and low fire incidence years. We first calculated the daily number of active fire pixels from 1 January 2001 to 31 December 2020 in the Eurasian Subarctic. Then we sorted these days in descending order according to the daily number of active fire pixels, and selected the first 20% of the days as high fire incidence days and the remaining 80% of the days as low fire incidence days. A year with at least one high fire incidence day is defined as a high fire incidence year, and a year without any high fire incidence days is defined as a low fire incidence year. Then, we define the fire season, and differentiate between peak and normal fire seasons. The fire season is the period that encompasses 90% of fire pixels for the year. We label these a peak fire season in the high fire incidence years, and normal fire seasons in low fire incidence years. Hayasaka et al. [21] used the term very active fire period to discuss the fire weather of high fire incidence years, which is similar with the peak fire season used in our work. However, they did not provide a quantitative method to define the very active fire period.

### 2.4. Generation of Fine Resolution Data

Our fine resolution data focuses on the dynamics of individual fires in the Eurasian Subarctic, as well as the adjacent region to the south. To analyze the development of individual fires, high spatial resolution remote sensing images are needed to track the perimeters of fires from ignition to extinction. For example, although the GlobFire database does include fire perimeters, the low spatial resolution results in some mixed fire instances and the loss of many small-scale fires. We, therefore, used Sentinel-2 L2A data, with a spatial resolution of 20 m, to develop a fire perimeter dataset to track the evolution of 719 individual fires (Figure 3). We selected those fire cases according to the spatial distribution of MCD14ML hot spots, which led them to be not very evenly distributed in the whole study area. We further processed the fire perimeter dataset to obtain the fire front dataset for analyzing the propagation characteristics of the fire fronts. Of these 719 fires, 442 fires were between 60 and 75°N in the main Eurasian Subarctic study area, and the remaining 277 were located between 50 and 60°N in the high-latitude interior of the continent, south of the main study area.

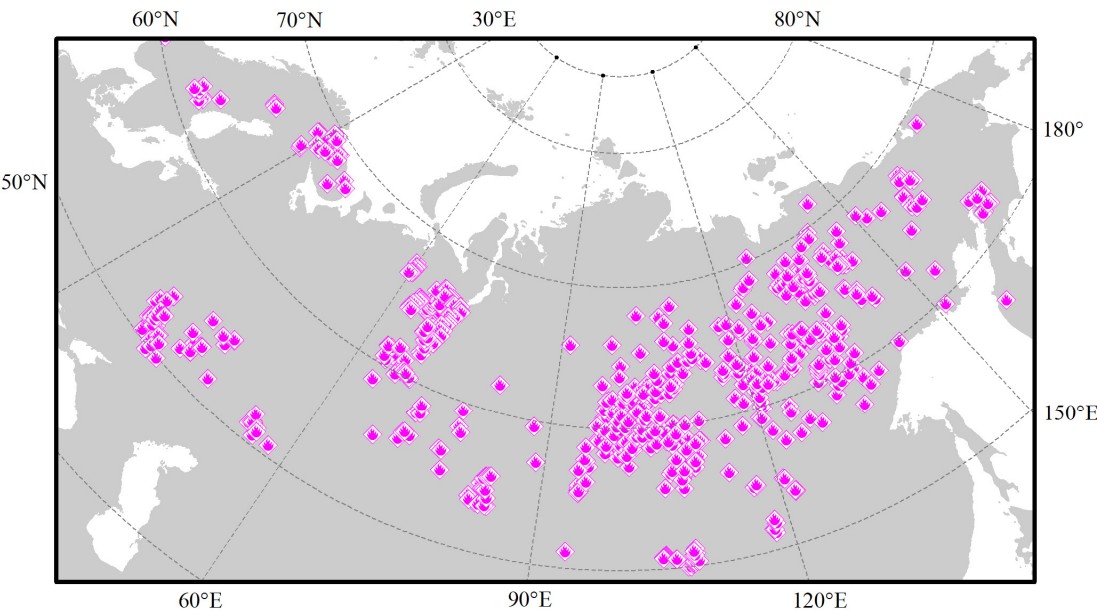

**Figure 3.** Spatial distribution of the 719 fires mapped (magenta diamonds). Of the 719 fires, 442 fires were in the region above 60°N, in the main study area, and the remaining 277 fires were located in the region below 60°N.

The observational time interval between fire perimeters was on average 5 days, though occasionally it was as short as 2 days; clouds were the principal factor limiting observations. We documented not only the location, time, and burnt area of each fire but also the daily expansion area, duration, propagation speed, and direction of fire fronts.

### 2.4.1. Fire Perimeter Dataset

The automatic extraction of the burnt land perimeter for our custom fire dataset uses spectral indices, region growing, and morphological processing. The overall procedure is shown schematically in Figure 4a. Based on the location and time information of fire pixels in the coarse resolution data for the period 2017 to 2021 (i.e., 5 years), 16,019 tiles of Sentinel-2 L2A product, each 5490 × 5490 pixels, were downloaded from the Copernicus Open Access Hub (https://scihub.copernicus.eu/dhus/#/home (accessed on 1 December 2021)). These tiles cover the study area and the adjacent high-latitude area. Some tiles were not suitable for fire perimeter extraction due to clouds or the tile only including many pixels that were not imaged. In addition, fires recorded in only a single time period were excluded. Finally, we manually checked all the downloaded tiles for data quality and obtained 2260 effective tiles to track the 719 fires.

We calculated a spectral index, the Normalized Burn Ratio 2 (NBR2; Equation (1)) using the Sentinel-2 data. NBR2 is a modified version of the Normalized Burn Ratio (NBR) [22] and uses only SWIR bands instead of a combination of NIR and SWIR bands. We empirically found NBR2 is more effective for post-fire area extraction in the Subarctic environment. NBR2 images from before ("pre") and after ("post") the fire are then subtracted (Equation (2)) to extract a preliminary map of burnt lands.

$$NBR_2 = \frac{B_{11} - B_{12}}{B_{11} + B_{12}} \tag{1}$$

$$dNBR_2 = -NBR_{2post} \tag{2}$$

$B$ represents a Sentinel-2 band, and the numeric subscript represents the band number ($B_{11}$ is centered at 1.61 μm and $B_{12}$ at 2.19 μm).

Because our interest is in tracking the burnt areas associated with actively burning fires, we screened all burnt lands identified using the $dNBR_2$ ratio for active fire pixels as indicated by the active fire detection index ($AFD_3$) [23].

$$AFD_3 = \frac{B_{12}}{B_{8a}} + \frac{B_{12}}{B_{11}} + \alpha \frac{B_{8a}}{B_{11}} \tag{3}$$

$\alpha$ is a scaling factor in the range [0, 1]; we used a value of 0.5 as recommended in [23]. As before, $B$ represents a Sentinel-2 band ($B_{8a}$ is centered at 0.86 μm). A threshold of 8.0 for identifying active fire pixels using the $AFD_3$ spectral index was chosen based on trial and error. This threshold was set deliberately high, and although this decreased the number of pixels labeled as active fire pixels, our algorithm only required a single $AFD_3$ active fire pixel to detect a fire.

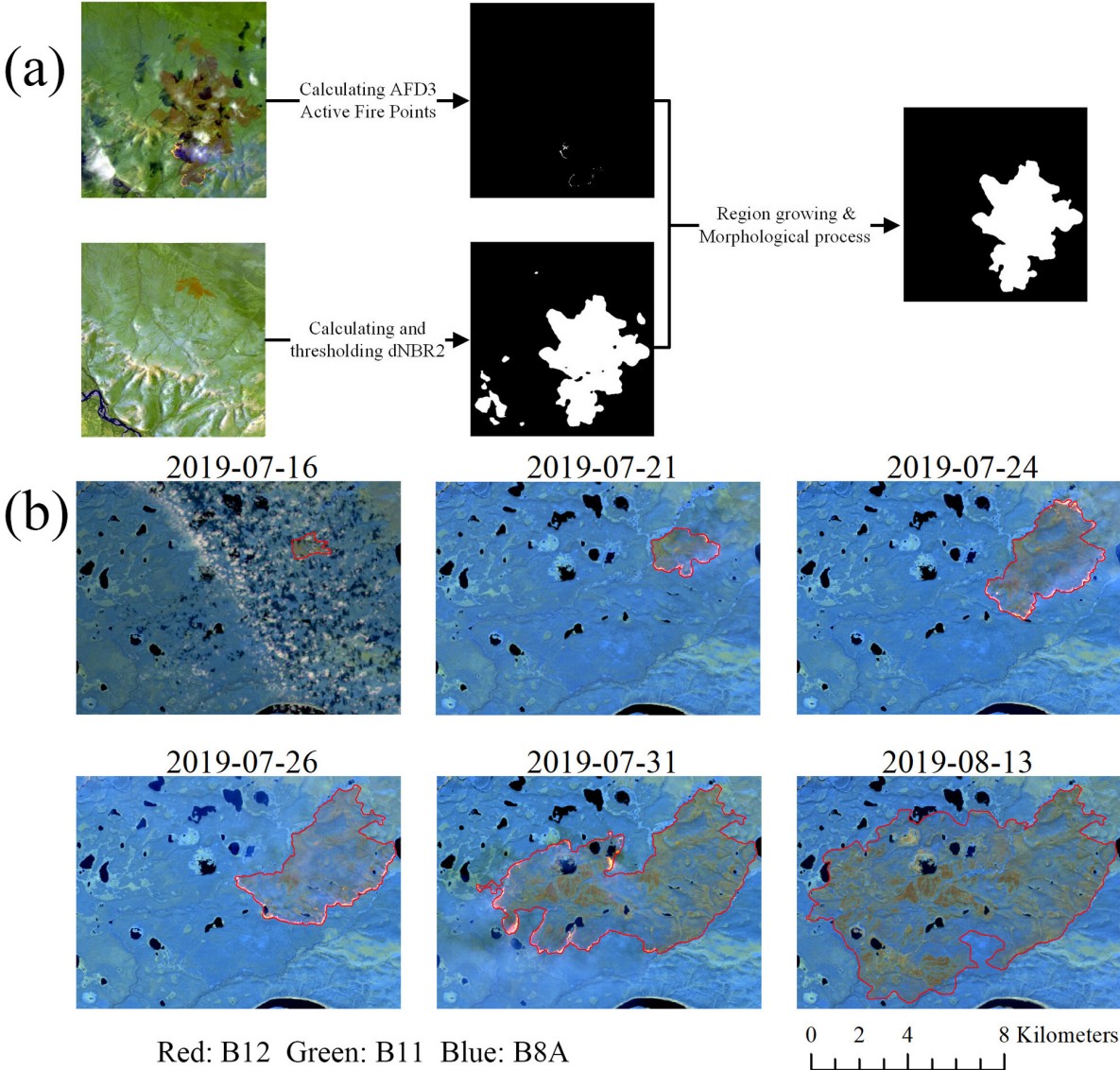

**Figure 4.** Tracking fires. (**a**) Schematic flow chart of burnt land entities extraction. Top row is an active fire image, while bottom row the corresponding image before ignition. (**b**) Tracking of a single fire, as shown by the red polygon. Bands 8a, 11, and 12 of the Sentinel-2 L2A product were used to extract the perimeters of the burnt land automatically.

The extracted active fire pixels were used as seeds to execute region growing segmentation on the dNBR2 image to produce a detailed map of the burnt land areas in each image. The perimeters of burnt lands were generated using a raster-to-polygon transformation algorithm. Morphological processing was applied, in which holes within polygons were filled (we use the binary_fill_holes operation in the Scipy mathematical algorithms collection [24]), and the edges of burnt lands were smoothed by applying a 5 × 5 median filter. This smoothing operation preserves holes with a large size because they usually correspond to unburnt areas. Finally, a manual check of the perimeters was carried out to ensure the accuracy of the extraction. This was the most time-consuming task in the production of the fire front dataset. Figure 4b shows an example of the development of a single fire from our fire product. Once we obtained the sequential perimeters of a fire and its development timeline, the burnt area for each perimeter was easily calculated. The duration of the fire is the difference between the observation times of the last and first perimeters in the sequence. The geometric center of the first burn perimeter was used to estimate the location of ignition. Finally, the daily expansion area was derived from the burnt area and duration.

### 2.4.2. Fire Front Dataset

To characterize the evolution of fire fronts, we employed two-dimensional fire front vectors (FFV), as shown in Figure 5b. Information attached to each FFV includes the position of the fire front, observation time of the starting point, and the propagation duration, direction, and distance. The two endpoints of an FFV represent a fire front position observed, respectively, at time $t$ and $t + 1$, and the direction of the FFV represents the fire front propagation direction. The endpoints of FFVs are generated by uniform sampling on the fire front observed at time $t + 1$. Each sampling point is used as the anchor to search for a suitable starting point for the associated FFV on the fire front observed at time $t$. The search rule used the direction from the anchor to the initial ignition location. However, for some geometries, this rule results in FFVs that would cross unburnt land or land previously burnt. For example, in Figure 5a, there is an unburnt area at the bottom of the map (i.e., southern side, the area almost entirely enclosed by red and dark red colors). Similarly, the fire front in top of Figure 5a has curled back towards the ignition point on 31 July. For both these situations, the search rule using a direction back to the fire ignition point will result in inappropriate FFVs. For these situations, the search rule switches to the shortest line between the current sampling point of fire front at time $t + 1$ and fire front at $t$, without the constraint that the direction should be towards the initial ignition.

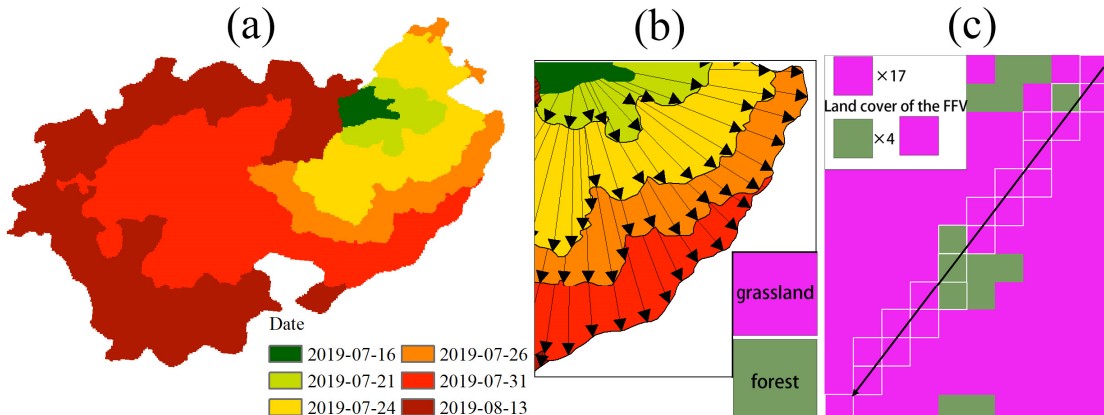

**Figure 5.** Fire front vectors. (**a**) An example of the progression of a single fire and (**b**) the associated fire front vectors for a portion of the fire. (**c**) Determination of land-cover type of an FFV. In this example, the FFV is assigned to grassland.

Using this process, 128,881 FFVs in total were generated for the individual fires. The 442 mapped fires in the Eurasian Subarctic were used to produce 107,613 FFVs, and the 227 fires of the adjacent region (below 60°N) produced 21,268 FFVs. Regions A, B, and C had 12,433, 76,766, and 18,414 FFVs, respectively.

To explore propagation characteristics of fire fronts on different land-cover types, we overlaid the FFVs on the ESA Land Cover CCI Product [20]. We tabulated the number of pixels for each land cover code through which the FFV passes, and the FFV was assigned the type of the land cover with the highest frequency, as shown in Figure 5c. Although this necessarily suppresses minor classes, it provides a simple method of determining the dominant land cover for each progression along the perimeter of the fire.

## 3. Results

### 3.1. Temporal Trends in Fires

This study focuses on the first two decades of the current century (2001 to 2021), coinciding with the period for which detailed satellite data is available. Because these data are only available for two decades (or less, for some datasets), it is important to acknowledge the consequent limitations on the robustness of interpretations regarding fundamental changes in fire occurrence. Nevertheless, Figure 6a, which places the study period in a longer, 40-year context, shows that the recent two decades were a period of dramatic change in burnt area in the Eurasian Subarctic, compared with a relatively consistent, low annual burned area before 2000 (source: FireCCILT11 [25]). After 2000, the annual burnt area varied greatly from year to year and between regions. This is explored in more detail in the remaining part of the figure. Figure 6b–d summarize the time-series of annual fire frequency (number of MODIS fire pixels [16]), burnt area [18], and carbon emissions from biomass burning [17] from 2001 to 2021. The main trends that can be seen in these graphs are discussed below.

1. In region A, which covers the East European and West Siberian Plain, the fire frequency, burnt area, and carbon emissions from biomass burning show perennial low fire occurrence, with the largest peak in 2012. Region A comprises 40% of the Eurasian Subarctic land area and is larger than regions B and C but, nevertheless, only ac-counts for a minority of the fires: 10% of the total number of fire pixels, 9% of the burnt area, and 8% of carbon emissions from biomass burning.

2. In region B, which covers the Central Siberian Plateau, the patterns of fire frequency, burnt area, and carbon emissions from biomass burning appear to have changed from 2012 onwards. With the exception of 2001 and 2002, fire frequency, burnt area, and carbon emissions from biomass burning were low prior to 2012. From 2012 onward, fire metrics were consistently high each year, with the only exception being 2015. Fire frequency and carbon emissions both suggest rising fire occurrence in recent years with unusually high peaks in 2021. The 2021 peak in carbon emissions from biomass burning is particularly large, increasing by 3.6 times compared to that of 2020, when the fire frequency increased by only 1.9 times. This increase could be caused by fires that were more intense (i.e., burnt more of the available fuels), or burnt land with larger carbon reservoirs [26]. The median values of fire frequency, burnt area, and carbon emissions from biomass burning between 2012 and 2021 are 5.7 times, 4.6 times, and 5.3 times the equivalent values for the period between 2001 and 2011. Overall, these statistics suggest a switch to higher fire occurrence, with more frequent fires, larger burnt area, and increased carbon emissions from 2012 onwards for this region.

3. For region C, which covers the East Siberian Highlands, the fire frequency, burnt area, and biomass burning carbon emissions show a marked periodicity. The time series is characterized by a majority of years with relatively low fire frequency, burnt area, and carbon emissions, punctuated by intermittent years with unusually high values in 2003, 2010, and 2020. The preceding 1–2 years before each peak in the active fire numbers are also typically elevated, though substantially lower than the peak.

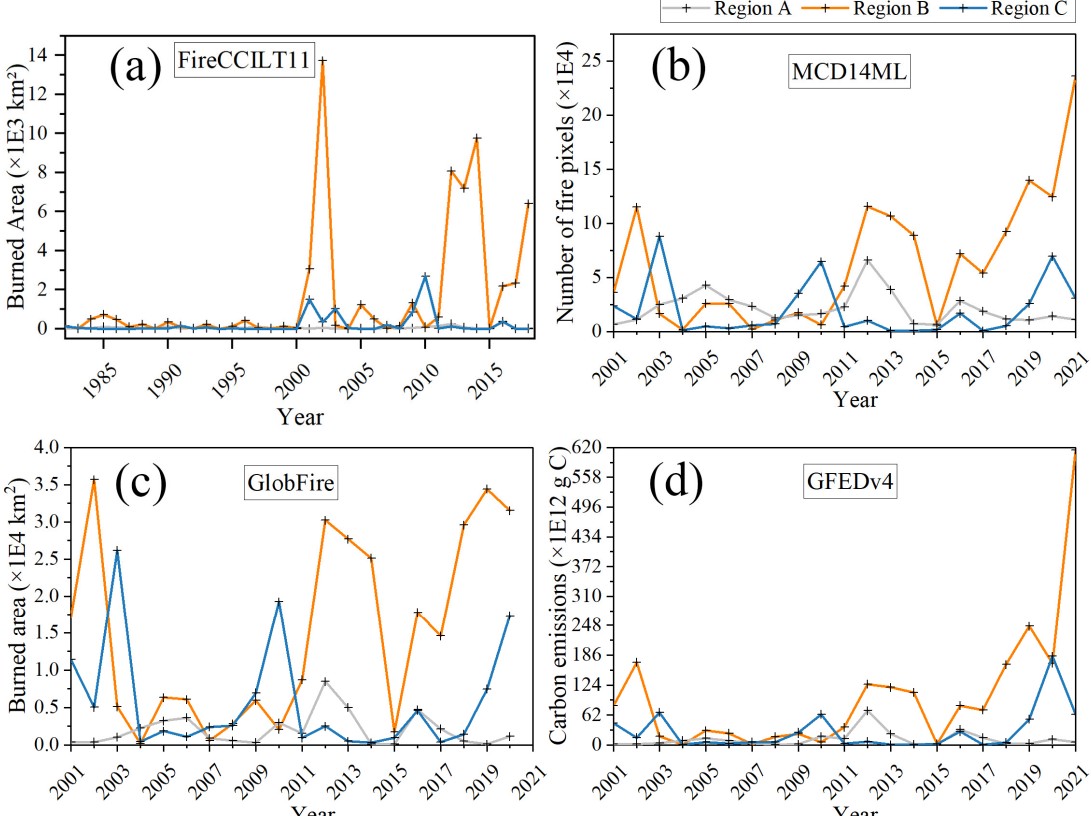

**Figure 6.** Temporal trends of fires based on number of fire pixels, burnt area, and biomass burning carbon emissions in the three regions of the Eurasian Subarctic. (**a**) Annual burnt area from 1982 to 2018. (**b**) Annual number of MODIS fire pixels from 2001 to 2021. (**c**) Annual burnt area from 2001 to 2020. (**d**) Annual carbon emissions from burning from 2001 to 2021.

### 3.2. Latitudinal and Longitudinal Trends in Fire Incidence

Figure 7a, derived from the MCD14ML data [16], explores the temporal evolution from 2001 to 2020 of fires within longitudinal bands. The number of active fire points is variable along the longitudinal gradient with West Siberian Plain (region A) exhibiting the lowest value, the Central Siberian Plateau (region B) showing the highest value, and the East Siberian Highlands (region C) showing more interannual variability in number of active fire points (Figure 7a). The figure also shows that the boundary between the regions A and B is gradational, with fire generally more frequent east of ~60° compared to the rest of region A. In Figure 7c, the second-order polynomial curve for the locations of the whiskers representing 90% of fire pixels for each year (red dashed line) indicates that the northern limit of most fires in the Eurasian Subarctic was relatively stable between 2001 and 2012 at approximately 66.0° but then moved northward gradually from 2013 onwards. These interpretations were evaluated using null-hypothesis significance testing. The slope of the trend line of 90% whiskers between 2001 and 2012 is not significantly different from zero (*p*-value: 0.34), and the slope of the trend line of 90% whiskers between 2013 and 2020 is significantly different from zero (*p*-value: 0.03). Furthermore, the median latitude of the northern limit of fires between 2013 and 2020 is 1.5° higher than it was between 2001 and 2012 (67.5° vs. 66.0°N). In Figure 7b, the temporal evolution of the median summer air temperature at 2 m above the ground surface (T2M) of the 30 latitudinal bands shows a similar trend of increasing values at high latitude. Indeed, the summer T2M and the number of fire pixels for the region between 60 and 75°N has a correlation coefficient of 0.6.

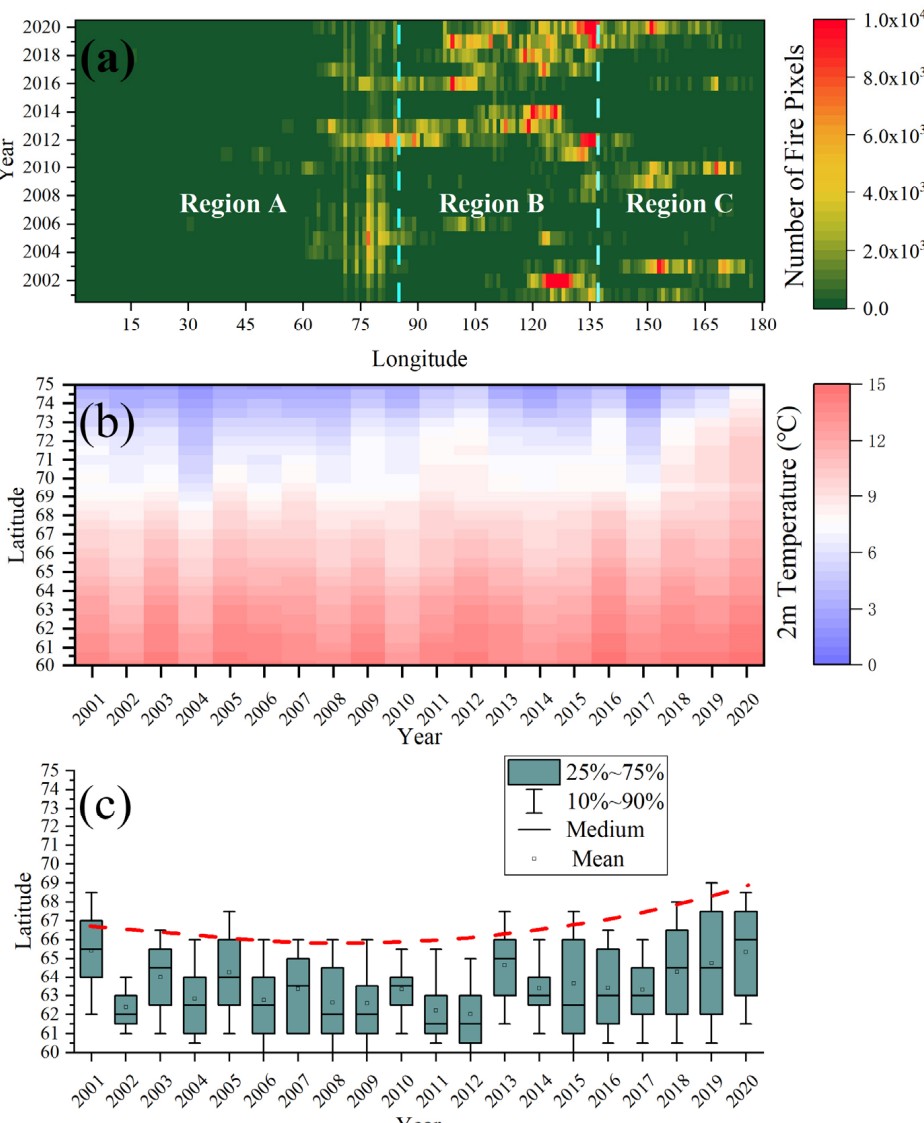

**Figure 7.** Summary trends in the spatial distribution of MCD14ML active fire pixels. (**a**) Longitude vs. time. (**b**) Latitude vs. time, for median summer temperature of air at 2 m above the surface (T2M) of land. (**c**) Boxplot for latitudinal distribution of fires over 20 years. Red dashed line indicates the second-order polynomial curve for whiskers representing 90% of the fire pixels for the specified year.

### *3.3. Fire Season Trends*

The daily dynamics of active fires from 1 January 2001 to 31 December 2020 in the Eurasian Subarctic is shown in Figure 8a. The figure shows that, as expected, fire incidence is highly variable, both within and between years. To facilitate discussion, we distinguish high fire incidence years as having one or more days with the number of fire pixels in the pink zone in Figure 8a (see methods section for more detail on how this threshold is set). The figure shows that Regions A and C had, respectively, only 5 and 4 high fire incidence years during the past two decades, but region B had 9 high incidence years, 8 of which occurred in the second decade, after 2011.

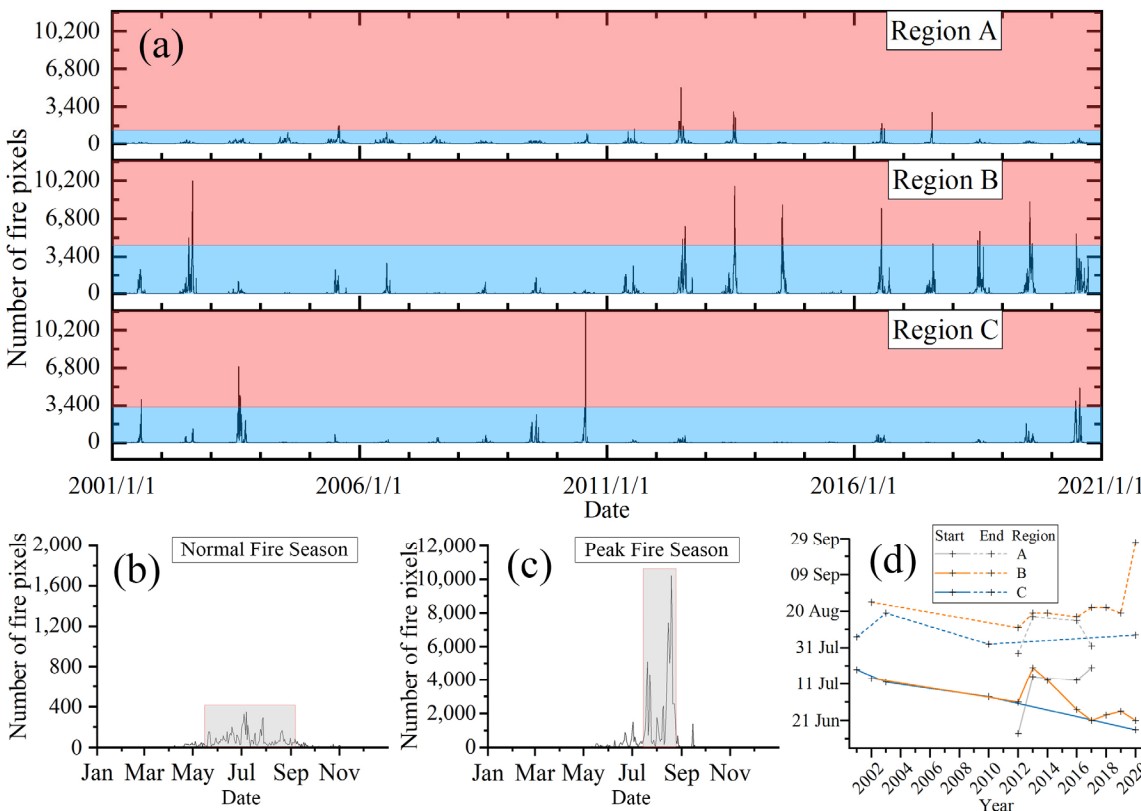

**Figure 8.** Fire seasons and associated terminology used in this paper. (**a**) Definition of "high fire incidence year". (**b**) Example of "normal fire season" in region A and (**c**) "peak fire season" for region B, as indicated by the gray shading. (**d**) The start and end dates of peak fire seasons for high fire incidence years in three regions.

For convenience, we also distinguish peak fire season in a high fire incidence year as the region shown as the gray shaded box in Figure 8c (see methods section for further detail on how this region is defined). Peak fire seasons usually have a shorter period duration than normal fire seasons in low fire incidence years (shown as the gray shaded box in Figure 8b); the latter is typically five months long, from May to September. Figure 8d shows the start and end dates of peak fire season for high fire incidence years in the three regions. Peak fire season in the three regions generally starts between late June and mid-July. The start dates of peak fire season in regions B and C advanced (i.e., are earlier in the year) at the rate of 1.40 and 1.64 days per year, respectively, with *p*-values from null-hypothesis significance testing (H0: the slope of the trend line is equal to zero) of 0.04 and 0.01, respectively. Region B is notable for the longer duration of the peak fire season, 54 days on average, compared to, respectively, 32 days and 33 days for regions A and C. In addition, in Region B, there is a general trend of a longer period of peak fire in recent years, with an earlier start time and later end time. According to the daily dynamics of active fires (Figure 8a), the number of active fire pixels generally peaked in July, agreeing with the results of Valendik [27].

### 3.4. Fires on Different Land-Cover Types

Burnt lands are mainly concentrated in the region from 90°E to 180°E, which generally coincides with the Central Siberian Plateau and the East Siberian Highlands. Fires tend to be distributed throughout the area from 60°N to 70°N, though fires are rare at the extreme northern margin of the study area. By overlapping the GlobFire Burnt Area Product [19] and the ESA Land Cover CCI Product [20], we documented the total burnt area over the 20 years studied in Table 2, for each of the five major land-cover types. We reclassified land-cover types in ESA Land Cover CCI Product [20], considering the classification

quality of mosaic landscapes in high latitudes. Tundra and grassland were merged to tundra; scrubland and cropland were merged to scrubland/herbaceous; forest and wetland remained unchanged. The rest of the land-cover types were merged to others. The spatial and temporal distribution of the burnt area is shown in Figure 9. According to Table 2, the overwhelming majority of the burnt area is forest land (87.1%), and forest is also the most common land cover burnt in each of the three regions. Burnt forests contributed 79.3% of carbon emissions from biomass burning in the whole Eurasian Subarctic, which is consistent with the estimates of Walker et al. [28] and Veraverbeke et al. [29]. Because region B had the most fires, it contributed the majority of the burnt forest (71.9%). Scrubland/herbaceous, tundra, and wetland account, respectively, for only a further 3.9%, 3.3%, and 2.6% of the burnt area for the region as a whole. However, wetland fires are concentrated in region A (where 57.0% of burnt wetlands were found). Scrubland/herbaceous fires were widely distributed throughout regions B and C. Region C is where 83.1% of burnt tundra is found.

**Table 2.** 20-year total burnt area for each of the five main land-cover types. The other land-cover classes include non-vegetated classes of urban areas, bare areas, and areas labeled as water bodies. "Type/A (%)" represents the area burnt for each land-cover type as a proportion of the area of region A. "A/ALL (%)" represents, for the total burnt area of the specified land-cover type, the proportion contributed by region A.

|  | Forest | Wetland | Scrubland/Herbaceous | Tundra | Others | Total |
|---|---|---|---|---|---|---|
| Total burnt area (km$^2$) | 397,614 | 12,025 | 17,426 | 15,160 | 14,084 | 456,309 |
| Proportion of total (%) | 87.1 | 2.6 | 3.9 | 3.3 | 3.1 | 100.0 |
| Type/A (%) | 68.1 | 17.4 | 7.8 | 2.4 | 4.3 | 100.0 |
| Type/B (%) | 94.2 | 1.2 | 2.4 | 0.6 | 1.6 | 100.0 |
| Type/C (%) | 74.8 | 1.3 | 6.0 | 11.2 | 6.7 | 100.0 |
| A/ALL (%) | 6.7 | 57.0 | 17.7 | 6.4 | 12.0 | - |
| B/ALL (%) | 71.9 | 30.5 | 42.8 | 10.5 | 33.9 | - |
| C/ALL (%) | 21.4 | 12.5 | 39.5 | 83.1 | 54.1 | - |
| Total | 100.0 | 100.0 | 100.0 | 100.0 | 100.0 | - |

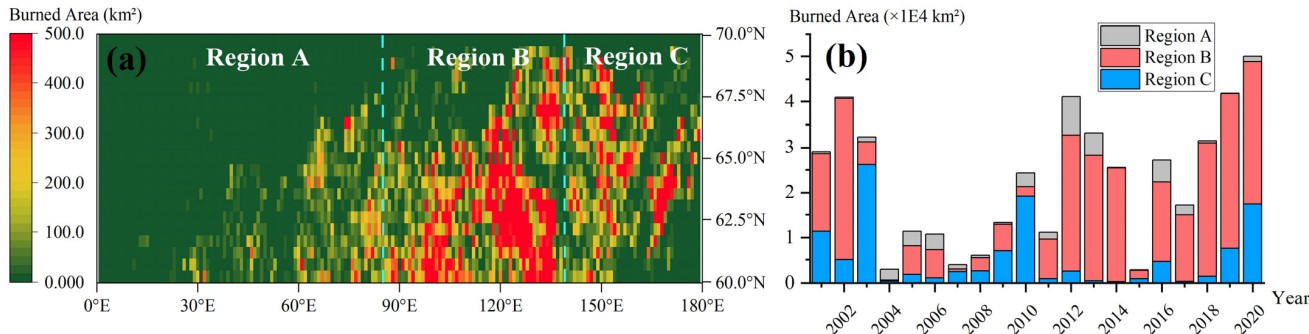

**Figure 9.** Spatial and temporal distribution of burnt area. The spatial distribution of burnt area over 20 years is shown in (**a**). The temporal distribution of burnt area in three regions and their contribution to the total burnt area in the whole Eurasian Arctic is shown in (**b**).

*3.5. Propagation Characteristics of Individual Fires*

In this section, we explore the characteristics of individual fires (442 fires in Eurasian Subarctic and 277 fires in the adjacent lower latitude regions, for the period 2017–2021) from the high-resolution remote sensing data. A key concept for this analysis is fire front vectors (FFVs), which represent the direction and distance that the fire boundary moved between satellite observations. The results show that fires in the Eurasian Subarctic tend to expand more rapidly and last longer than those in the region immediately to the south.

Particularly, fires in the East Siberian Highlands (region C) grew the most rapidly, and fires in the Central Siberian Plateau (region B) lasted the longest.

As shown in Table 3, the average duration of the 442 fires in the Eurasian Subarctic study area was 13 days, and the average daily expansion in the area was 7.5 km$^2$/d. The daily area of expansion is defined as the newly burnt surface area after one day and is obtained by calculating the area difference between the fire perimeters on different days of the timeline (see methods section for further detail). In comparison, the average duration of the 277 fires in the adjacent region (i.e., 50 to 60°N) was much shorter, only 8 days, and the average daily expansion area was only 3.0 km$^2$/d. Fire fronts in the Eurasian Subarctic had a median propagation speed of 446 m/d, compared to 280 m/d in the adjacent region to the south. While the fires in the Eurasian Subarctic have higher expansion speed and longer duration compared to the south regions adjacent to the subarctic, the propagation of fire fronts within the three regions of Eurasian Subarctic also show marked differences (Table 3). Region B had notably longer duration fires (18 days vs. 6 and 10 days for the other two regions), whereas fires in Region C grew much more rapidly (14.1 km$^2$/d vs. 2.6 and 8.6 km$^2$/d for the other two regions).

**Table 3.** Propagation speed, direction, and duration of individual fires.

| Region | Duration (Days) | Daily Area of Expansion (km$^2$/d) | Propagation Speed of Fire Fronts (m/d) | Main Propagation Direction of Fire Fronts |
|---|---|---|---|---|
| All (60–75°N) | 13 | 7.5 | 442 | West and NE |
| A | 6 | 2.6 | 464 | NW |
| B | 18 | 8.6 | 388 | West and NE |
| C | 10 | 14.1 | 832 | SW, NW and NE |
| Adj. region (50–60°N) | 8 | 3.0 | 280 | SW and NE |

Fire fronts in the Eurasian Subarctic propagated more commonly towards the west and northeast, whereas in the south adjacent region, southwestward-propagating fire fronts were more common. In region A, fire fronts mostly propagated towards the northwest. The pattern of propagation directions in region B is very similar to that of the Eurasian Subarctic as a whole, with westward and northeastward-propagating fire fronts. This is because region B has the most fires in the Eurasian Subarctic and, therefore, dominates the aggregate summary. Fire fronts in region C propagated mainly towards the southwest, northwest, and northeast directions.

## 4. Discussion

### 4.1. Factors Affecting Fire Patterns

The initiation of wildfire has three main components: an ignition source, fuel (i.e., vegetative biomass), and environmental conditions that promote combustion [30]. Once a fire has ignited, propagation of the fire is also dependent on topography and landscape structure. Weather can affect both ignition sources (e.g., lightning) and fuel combustibility. Below, we explore three key factors of weather, land cover, and landscape structure.

#### 4.1.1. Weather

Weather conditions, such as humidity, temperature, and wind, are a key factor affecting the development of fires. The relationship between extreme fire activity and weather is well established. For example, extreme fire weather is often associated with a decrease in relative humidity and an increase in temperature [31].

We collected annual median summer FWI (fire weather index, where higher values represent higher risk of fire) from ERA5 reanalysis [32], annual median summer precipitation (P) from GPCP monthly mean precipitation rates v2.3 [33], and annual median

summer temperature of air at 2 m above the surface of land (T2M) from ERA5-Land [34]. Their correlations with fire frequency (FF), burnt area (BA), and carbon emissions (CE) are shown in Figure 10a–c. FWI shows much higher correlation coefficients with FF, BA, CE in regions B (Figure 10b) and C (Figure 10c) compared to region A (Figure 10a), where the correlation coefficients are close to zero. The correlation coefficients of T2M-FF, T2M-BA and T2M-CE in region B are higher than that in regions A and C. Notably, in regions B and C, P-FF, P-BA, and P-CE show the expected negative correlation, but in region A, they are positively correlated.

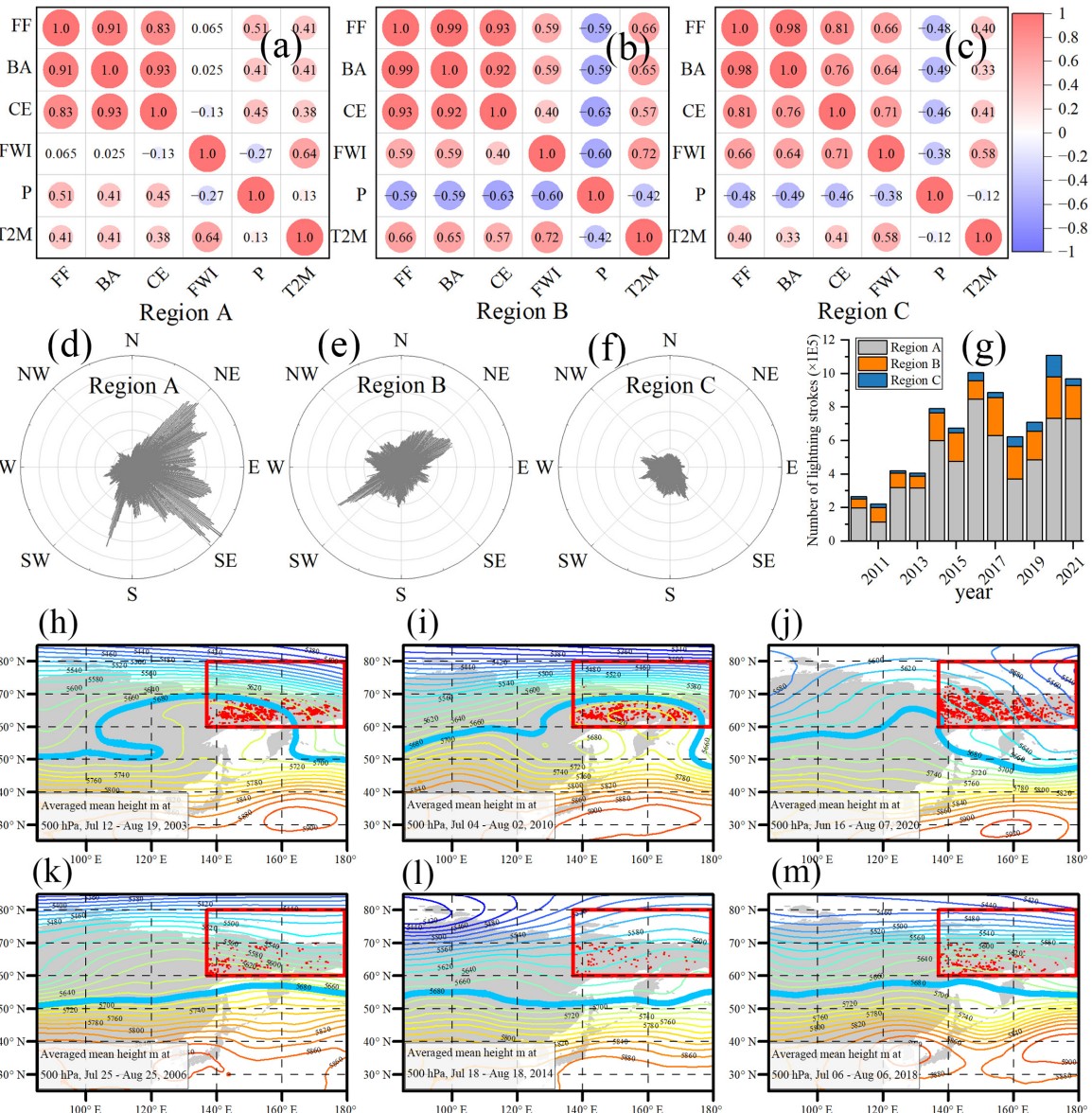

**Figure 10.** Weather factors affecting patterns of fires. (**a–c**) Correlation coefficient maps among FF (fire frequency, source: MCD14ML), BA (burnt area, source: GlobFire), CE (carbon emissions, source: GFEDv4), FWI (fire weather index, source: ERA5 reanalysis), P (precipitation, source: GPCP monthly mean precipitation rates v2.3), T2M (2 m temperature, source: ERA5-Land) in three regions, A, B and C. (**d–f**) Wind roses for the three regions (June-October). (**g**) The annual number of lightning strokes in the three regions from 2010 to 2021. (**h–m**) Averaged weather maps at upper air (500 hPa) in fire season. The isoline (5680 m) is thickened to highlight the pattern of the westerlies. The red rectangle represents the extent of region C. Red pixels represent fire pixels. (**h–j**) High fire incidence years (2003, 2010, 2020). (**k–m**) Low fire incidence years (2006, 2014, 2018).

Our results of the fire propagation characteristics indicate the selectivity of propagation direction of fires, which is affected by wind direction [35] and local landscape structure. The main wind directions (Figure 10d–f) and the propagation directions of fire fronts in regions A, B, and C (Table 3) are similar in distribution.

The marked periodicity of fire in region C may be driven by large meanders in the westerlies. Figure 10h–m shows averaged weather maps at upper air (500 hPa), and the periods cover peak fire season for high fire incidence years and a month in the fire season for low fire incidence years (source: ERA5 hourly data on pressure levels [36]). Meanders in the westerlies bring warm air masses to high latitude regions, creating fire-favorable conditions [21]. Fire size is sensitive to weather in the days to weeks following ignition [37]. The presence of warm air mass and persistent high-pressure systems in the upper air are strongly associated with large-scale fires in the Subarctic [38]. The averaged weather maps Figure 10h–j show persistent meandering of westerlies during the peak fire seasons of high fire incidence years, unlike the low fire incidence years, as shown in Figure 10k–m.

The ratio of lightning strokes in the Subarctic to the number of global strokes has increased between 2010 and 2020, and this increase appears to be correlated with the increasing temperatures in the Subarctic [39]. As shown in Figure 10g, the trend of increasing number of strokes in the Eurasian Subarctic is notable (source: WGLC [40]) and can be observed in all three regions. The number of observed strokes is greatest in region A and the smallest in region C.

### 4.1.2. Land Cover

Land cover clearly has a major influence on fire patterns [10,11,41,42]. Previous studies [43,44] have shown that shrublands are more fire prone, while agricultural areas are less fire prone. In our work, the importance of combustible materials is illustrated by the results of the medium and fine resolution analyses (Table 2). Furthermore, the speed of the propagation of the fire fronts varies by land-cover type. Fire fronts on tundra had the fastest median propagation speed, 505 m/d, followed by forest and scrubland/herbaceous, 405 m/d and 378 m/d, respectively. Fire fronts on wetland have a relatively slow median propagation speed, 344 m/d. Furthermore, the distribution of land-cover types differs between the regions, and thus, the regional difference in the proportions of combustible materials contributes to the different patterns of regions.

### 4.1.3. Landscape Structure

Landscape structure, including local topography, boundaries of farmland, and hydrography, also affect fire propagation. The study by Povak et al. [45] documents how topographic features, including both valleys and ridges, as well as roads, influence the spatial growth of fires. Land cover that is slow to burn or acts as a firebreak, such as farmland [43] (in certain seasons), wetland, rivers, and lakes, may constrain the propagation and the resulting footprint of fires.

Figure 11 provides some typical examples of individual fire propagations. Some of them are restricted and shaped by the boundaries of waterbody or topography, the others in more homogenous landscapes often spread in all directions, which tends to result in burnt areas that are more compact in shape and potentially large in area.

### 4.2. Future Fire Risks in the Eurasian Subarctic

The characteristics of fire in the Eurasian Subarctic differs markedly from the adjacent region to the south, with longer duration, larger daily expansion area, and faster propagation (Table 3). The reason for these differences may also lie in the differences in the proportions of combustible materials, as well as terrain and human activities, especially the fire management [46] and the weather conditions in the days to weeks following ignition. The Subarctic region is characterized by extensive areas of peat, which can burn for long periods when these fuels ignite [13,47]. Although greater human population density in the southern adjacent region may favor an increase in the number of fires, timely efforts

to extinguish fires and firebreaks created by farmland, roads and buildings may limit expansion of fires.

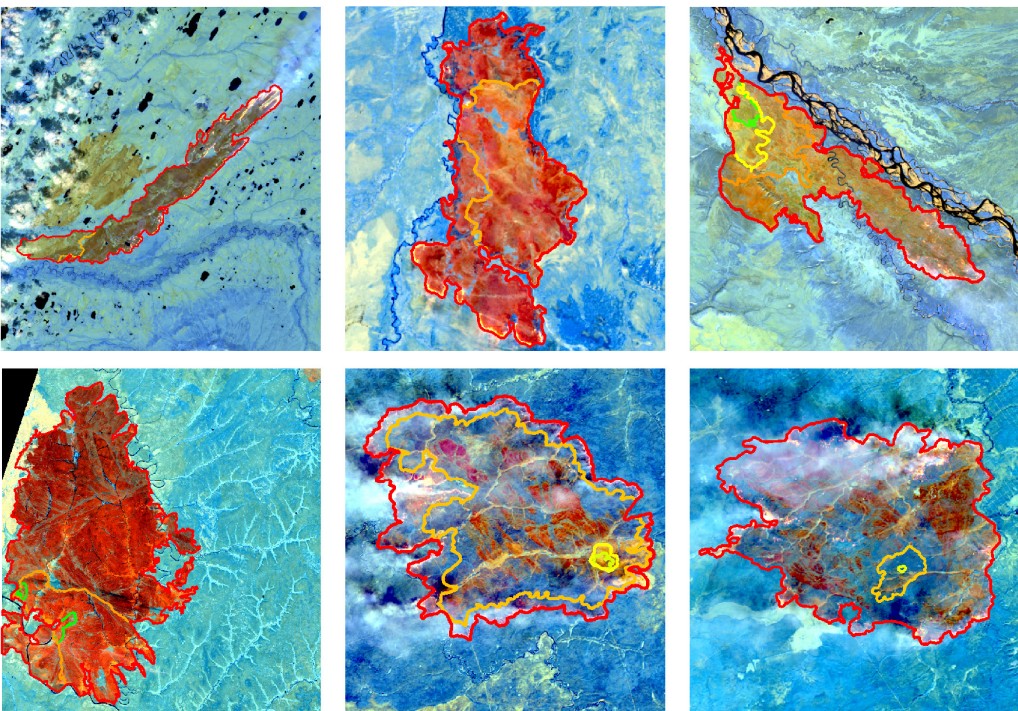

**Figure 11.** Examples of typical individual fires on different landscape structures. The images are Sentinel-2 bands B12, B11, B8A as RGB; the polygons of different colors represent successive fire boundaries as mapped by our algorithm.

Looking forward to the future of Eurasian Subarctic fire risks, empirical data and climate models indicate that high latitudes will be particularly affected by global climate change [48]. Modeling studies indicate a likely increase in fire weather severity, which may prompt changes in fire regimes [49]. Our results indicate that the T2M temperature in high latitude regions has increased with time. Furthermore, high latitude regions are likely to continue becoming warmer and drier [50], thus, potentially further increasing the risk of fire. Therefore, the increasing fire occurrence observed in the Eurasian Subarctic since 2012, dominated by region B, is likely to continue into the future, which agrees with the studies of McCarty et al. [1,51].

Our results indicate that the northern limit of fire is expanding poleward, especially after 2012, and that fire is becoming more frequent. This is the result of increasingly fire-favorable conditions, such as warming air and drier fuels [50]. Our results also discover a lengthening fire season in region B, providing additional evidence supporting the findings of Kirillina et al. [15], who studied most of region B for the period 1996 to 2018. The observed fire season trends suggest that peak fire season in the Eurasian Subarctic, which currently comprises the two months of July and August, may grow to a total of three months and include late June and early September. The overall length of the fire season in the Eurasian Subarctic, dominated by region B, is expected to further increase, which is part of global fire season lengthening promoted by combined surface weather changes [52]. The advancing of the start dates of the fire seasons appears to be driven by the changing summer climate, and results from Sun et al. [53] show a significant positive correlation between summer land surface temperature and spring snowmelt. The study by Westerling [54] provides an example how reduced winter precipitation and an early spring snowmelt can prompt a shift from a short fire season to longer one.

The Subarctic is the region where warming associated with global climate change is the most evident in recorded measurements from the last 120 years [48]. Climate change will

likely lead to landscape changes in the high northern latitudes, such as tundra changing to forest [55] and forest–steppe to steppe [1]. Fires correlate with land-cover type, especially in the early stages, such as higher association with shrublands, pine stands and eucalypt plantations than with croplands. Such landscape changes, along with the increasing temperature and decreasing humidity in the Subarctic [30,56], will likely form more favorable conditions for the propagation of fires. Furthermore, climate-driven increases in lightning activity will likely provide additional ignition sources in the Subarctic permafrost [57].

## 5. Conclusions

In summary, this study of two decades of fire incidence, from 2000 to 2021 drew upon multiple types of massive remotely sensed data and derivative products, including the newly created one. These datasets offer insight into the geographically contrasting patterns of changing fire occurrence over the last two decades within the Eurasian Subarctic. We discover that the East European and West Siberian Plain (region A) is dominated by years of low fire incidence, with years of only occasional high incidence. In contrast, fire is generally more common in the Central Siberian Plateau (region B), the incidence of fire and fire carbon emissions have increased markedly since 2012. The East Siberian Highlands (region C) are notable for the apparent periodicity of high fire incidence and carbon emission years. Such regional differences in fire patterns were not reported before.

We also reveal that the length of the fire season in the Eurasian Subarctic, dominated by region B, is increasing with earlier start dates and may grow from two months to three months in duration. The northern limit of fire has extended approximately 1.5° to higher latitudes. The average duration, daily expansion, and propagation speed of individual fires are 13 days, 7.5 $km^2$/d, and 446 m/d, notably higher than the 8 days, 3.0 $km^2$/d, and 280 m/d in the region immediately to the south. The Eurasian Subarctic is dominated by forest land cover; forest is the main land cover burnt by fires, and forest produced 79.3% of the fire carbon emissions.

Our study indicates that the trends of increasing fires and fire carbon emissions in the Eurasian Subarctic are likely to continue or be reinforced in future years. Appropriate policies and measures should be formulated, but these responses should take into account the large geographical differences in the fire trends within the region.

**Author Contributions:** Conceptualization, S.J.; data curation, Y.Z.; formal analysis, Y.Z.; funding acquisition, S.J.; investigation, Y.Z.; methodology, Y.Z.; project administration, S.J.; resources, T.A.W.; software, Y.Z.; supervision, S.J.; validation, Y.Z.; visualization, Y.Z.; writing—original draft, Y.Z.; writing—review and editing, S.J. and T.A.W. All authors have read and agreed to the published version of the manuscript.

**Funding:** This work was supported by the National Natural Science Foundation of China (grant No. 42171430) and the State Key Program of the National Natural Science Foundation of China (grant No. 42030102).

**Data Availability Statement:** The fire perimeter and fire front datasets generated and analyzed during the current study are available at https://doi.org/10.6084/m9.figshare.20286561 (accessed on 11 July 2022).

**Acknowledgments:** We acknowledge the use of data from NASA's Fire Information for Resource Management System (FIRMS) (https://earthdata.nasa.gov/firms, accessed on 11 July 2022), part of NASA's Earth Observing System Data and Information System (EOSDIS).

**Conflicts of Interest:** The authors declare no conflict of interest.

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
