# Peer review of "Regional Spatiotemporal Patterns of Fire in the Eurasian Subarctic Based on Satellite Imagery"

_remotesensing, doi:10.3390/rs14246200_

Round 1

Reviewer 1 Report

The authors present a detailed analysis on spatial and temporal patterns of fire events in a region where fire risk is constantly growing. The amount of remote sensing and geospatial data that was used is quite large, however the authors manage to present a thorough and quantitative assessment of fire characteristics and at the same time preserve clarity in the manuscript text and structure. I do not have any important criticism on the research design and quality of presentation. I believe the findings are quite useful for the readers and the community.

Some minor comments here: 

 - What was the reason to choose the 2017-2021 period regarding the fine resolution analysis? An additional comment could be provided here.

- Figure 7 images numbers b and c do not correspond to captions. Please correct also in text these references.

Author Response

  1. What was the reason to choose the 2017-2021 period regarding the fine resolution analysis? An additional comment could be provided here.

Answer: Thank you. We revised the corresponding sentences to explain why we made such a selection of fire cases.

We selected those fire cases according to the spatial distribution of MCD14ML hot spots, which led them not very evenly distributed in the whole study area.

  1. Figure 7 images numbers b and c do not correspond to captions. Please correct also in text these references.

Answer: Thank you. Corrected.

Reviewer 2 Report

The manuscript by Yikang Zhou et al. “Regional spatiotemporal patterns of fire in the Eurasian Arctic … ” presents a very stimulating study conducted to better characterize spatial and temporal patterns of fire occurring over two decades period between 2001 and 2021 in the high-latitude terrestrial ecosystems of the Arctic Eurasian region.

The work exploited a large amount of information derived from satellite imagery concerning active fires, burnt areas and atmospheric emissions using remotely sensed data at three different level of spatial resolution from coarse to medium up to fine decametric resolution.

While for the coarse and medium resolution the investigation is based on publicly available fire products, for the scope of fine resolution analysis a novel methodology is proposed to derive a new fire related product which allows to track the spatial and temporal evolution of individual fires.

This fine resolution dataset (Fire Perimeter Dataset) generated from Senitnel-2 data for the years 2017-2021 covers also the adjacent lower latitude subarctic regions (50° -60° N); findings from this analysis highlighted  differences in the duration of individual fires and propagation speed of fire fronts in the arctic and subarctic regions.

The analysis was conducted for the entire study area of Eurasian Arctic region as whole, and interestingly  also for three sub-regions (East European and West Siberian Plain, 0°~85°E; Central Siberian Plateau, 85°~137°E; East Siberian Highlands, 137°~180°E), revealing for each sub-region specific fire patterns and relationship between fires and associated key factors such as land cover types, weather and landscape structure.

The study presented is interesting and the manuscript is well organized and clearly presented.

Minor comment:

The graph of T2M is showed as Fig 7(b) differently from the text (Line 293) “In Figure 7(c), the temporal evolution of the median summer air temperature at 2 m…” and figure caption (Line 301). Please rearrange

Author Response

  1. The graph of T2M is showed as Fig 7(b) differently from the text (Line 293) “In Figure 7(c), the temporal evolution of the median summer air temperature at 2 m…” and figure caption (Line 301). Please rearrange.

Answer: Thank you. The captions of Fig. 7b & 7c were rearranged to be consistent with the text.

Reviewer 3 Report

The research is very interesting, a large amount of data has been processed,  and very important and topical issues of the dynamics of fire development in high latitudes, which are generally difficult to access for field research, have been considered. However, there is a lack of consistency in the presentation of results, it may be better focusing on a few great results and presenting them more consistently. I have two main comments. The first refers to the name of the region. The region of the study is usually called Subarctic or Northern Taiga, since the Arctic is limited to the tundra and does not include forests. The second is methodological point. It should be noted that when comparing spatial data of different scales, errors accumulate and the reliability of the results and the conclusions drawn is doubtful. User guides and numerous studies indicate that the reliability of the satellite products used in the article drops significantly at high latitudes, and therefore, in addition to the built-in data quality fields, additional control masks, such as exclusion water and anthropogenic objects, should be applied. Below are comments, the answers to which may help clarify the details of the study and make its presentation ready to accept. 

95 The problem of terminology Arctic and Subarctic. And comparing with sourthen territories can be the theme for another research.

108 It seems to be wrong reference on MODIS MCD64A1 product.

111. Was the selection of MCD14ML data by Type field? If even yes, there is some obvious, non-fire static nature (volcanoes on Kamchatka, water bodies) or industrial hot-spot sources (for instance, a lot of gas flares on the north of A) are not properly flagged as such in the type field of the MCD14ML product. Usually it takes additional processing to remove these hotspots. If no, the overestimation will be even greater because of lot of inland water objects and open surface, which can give non-wildfire hotspots northern latitudes.

133 Was the union process made for every year or for the whole period?

136 ESA Land Cover CCI Product has a great limitation of using because of low quality classification of mosaic landscapes (water objects/shrubs/dense/open forest) in high latitudes. Such items like grassland and croplands are not usual for these regions. May be it is better to eliminate them to shrub-herbaceous vegetation or tundra.

164 Why was it necessary to select fires outside the regions, and on what basis were these 719 fires selected? It is not clear in this part of the article body.

299 Figure 7 - please indicate the source of pixels (What satellite data). 

326 The result is very interesting, but the graph is not very clear, may be you can try more efficient visualization of d)figure.

336 These cannot be cropland fires, since this is a zone of tundra, forest-tundra and northern taiga. There are industrial facilities on the Kola Peninsula that can give hotspots. Please check it or change the name of class to tundra, for instance.

337 Grassland it is more common for steep southern landscapes. If you look at the figure 1, you see that vast territories of grassland are situated in tundra close to the sea. It the high latitudes it is more herbaceous and shrubland mosaic with low dense forest or tundra. Therefore, in this case grassland and tundra is most likely synonyms than two different classes.

342 What is a permafrost fire? B and C regions are situated in permafrost zone. The area of this class is too small and within accuracy error, it may be better not to mention or classify like Others.

360-367 The conclusions are doubtful, since the land cover classification is unreliable. Perhaps, in this situation, it makes sense to combine data in several classes like –  1)tundra, 2)forest, 3)scrubland/herbaceous, 4) wetland  and 5)others (open ground, etc.) - since there are not wildfires, but rather anthropogenic object, glint or hot surface. Or maybe don’t discussed landcover in the article and more clearly describe other interesting results.

374 Strictly speaking, the region that is considered in the article is mainly the Subarctic, the Arctic is limited to the tundra, which has been burning only in recent years and to a very limited extent. Perhaps it would be more logical to limit this section to only 442 plots and describe the features of fire dynamics in sparse forests on permanent permafrost depending on latitude and longitude. Areas south of latitude 60 already are the taiga with dense forests, moistened swamps with partially unstable permafrost, and it is obvious that they will burn differently.

400 Perhaps, it is not necessary to single out subsections in the section, since LandCover raises questions, and LandStructure is rather descriptive, since there was no real study at least regions elevation.

441 References on researches in Portugal are unreliable, because it is another vegetation, soils and water regime in the classes then in Subarctic region.

452-453 Figure 11. It does not look like an cropland fire, but like a burned-out object, for example, a forested area or just a water body. Please check it. And if it is southern 60 latitude, it is not a good example.

 (c) Examples of fires which are affected by agricultural activities and hydrology (top row) – Need to be revised.

492 There is no such process in the region – “such as the northward expansion of agriculture” - this is a false statement.

494 “Fires correlates with land cover type, especially in the early stages [41], 495 such as higher association with shrublands, pine stands and eucalypt plantations than with croplands” – irrelevant reference.

495 “Our results indicate that fires on grasslands have the fastest propagation speed” – should be revised.

500 Conclusion should be revised.

Author Response

Reviewer 3:

  1. 95 The problem of terminology Arctic and Subarctic. And comparing with southern territories can be the theme for another research.

Answer: Thank you. We changed the expression of our research area from Arctic to Subarctic to be consistent with our research contents.

  1. 108 It seems to be wrong reference on MODIS MCD64A1 product.

Answer: Thank you. This wrong reference was revised.

  1. 111. Was the selection of MCD14ML data by Type field? If even yes, there is some obvious, non-fire static nature (volcanoes on Kamchatka, water bodies) or industrial hot-spot sources (for instance, a lot of gas flares on the north of A) are not properly flagged as such in the type field of the MCD14ML product. Usually it takes additional processing to remove these hotspots. If no, the overestimation will be even greater because of lot of inland water objects and open surface, which can give non-wildfire hotspots northern latitudes.

Answer: Yes, we excluded non-vegetation hot spots according to the value of the type field in the csv file (not equal to 0). Then, filtered out hot spots with low confidence value. Despite such processing, when we browse the remote sensing image corresponding to the fire points, we still find a few hot spots generated from factory towering chimneys. However, such non-vegetation hot spots are minority among the recorded ~100k hot spots every year in the 1 km MODIS Standard Fire product (in the scope of our research area), so we didn’t further deal with these non-vegetation hot spots.

  1. 133 Was the union process made for every year or for the whole period?

Answer: Thank you. Union process was made for all burnt area masks recorded from ignition to extinction of each fire case.

  1. 164 Why was it necessary to select fires outside the regions, and on what basis were these 719 fires selected? It is not clear in this part of the article body.

Answer: Thank you. We added sentences in the revised manuscript to explain why we made such selection.

Our fine resolution data focuses on the dynamics of individual fires in Eurasian Subarctic as well as the adjacent region to the south.

We selected those fire cases according to the spatial distribution of MCD14ML hot spots, which led them not very evenly distributed in the whole study area.

  1. 299 Figure 7 - please indicate the source of pixels (What satellite data).

Answer: Thank you. Done.

  1. 136 ESA Land Cover CCI Product has a great limitation of using because of low quality classification of mosaic landscapes (water objects/shrubs/dense/open forest) in high latitudes. Such items like grassland and croplands are not usual for these regions. May be it is better to eliminate them to shrub-herbaceous vegetation or tundra.
  2. 336 These cannot be cropland fires, since this is a zone of tundra, forest-tundra and northern taiga. There are industrial facilities on the Kola Peninsula that can give hotspots. Please check it or change the name of class to tundra, for instance.
  3. 337 Grassland it is more common for steep southern landscapes. If you look at the figure 1, you see that vast territories of grassland are situated in tundra close to the sea. It the high latitudes it is more herbaceous and shrubland mosaic with low dense forest or tundra. Therefore, in this case grassland and tundra is most likely synonyms than two different classes.
  4. 342 What is a permafrost fire? B and C regions are situated in permafrost zone. The area of this class is too small and within accuracy error, it may be better not to mention or classify like Others.
  5. 360-367 The conclusions are doubtful, since the land cover classification is unreliable. Perhaps, in this situation, it makes sense to combine data in several classes like – 1)tundra, 2)forest, 3)scrubland/herbaceous, 4) wetland  and 5)others (open ground, etc.) - since there are not wildfires, but rather anthropogenic object, glint or hot surface. Or maybe don’t discussed landcover in the article and more clearly describe other interesting results

Answer for 7-11: Thank you. According to the suggestions of the reviewer and the further investigation on the classification quality ESA Land Cover CCI Product, we focus our research on five types of land cover: forest, wetland, scrubland/herbaceous, tundra and others. We updated corresponding text and table in the revised manuscript.

  1. 492 There is no such process in the region – “such as the northward expansion of agriculture” - this is a false statement.

Answer: Thank you. Corrected.

  1. 494 “Fires correlates with land cover type, especially in the early stages [41], 495 such as higher association with shrublands, pine stands and eucalypt plantations than with croplands” – irrelevant reference.

Answer: Thank you. Corrected.

  1. 495 “Our results indicate that fires on grasslands have the fastest propagation speed” – should be revised.

Answer: Thank you. Done.

Reviewer 4 Report

The authors examined the fire variability in the area of ​​the Eurasian Arctic. Using publicly available databases, they made a set of calculations, dividing the research area into three regions. I think that the authors undertook quite an ambitious task. The research concept was pervasive, and the analyses provided much data that could easily create even two separate articles. The authors skillfully presented the entire research scheme to us based on selected examples and collective data. I think the article is correctly written and suitable for publication in Remote Sensing, mainly because of the interesting topic. I propose a few minor changes:
- slightly re-edit the introduction (add the clearly stated aim of the research),
- refine the figures,
- work on editing the text (different spacing between lines, spaces, dots, and other punctuation marks are missing in some places).
I added some minor comments in the file.

Author Response

Reviewer 4:

  1. Line 77-80 I think it would be good to add the area (e.g. in km2) of each region (A, B, C).

Answer: Thank you. Done.

  1. Fig. 1 The description of meridians and parallels is not clear (not intuitive). I suggest choosing a different font (one for meridians and the other one for parallels) or orienting the captions differently - e.g. captions of meridians parallelly to the meridians.

What is the data source?

Answer: Thank you. we use different font of captions (Italic and Regular) to make a clear description of meridians and parallels. Data source used in Fig. 1 is added to revised manuscript.

  1. Line 84-95 This part sounds like a summary. The introduction is not a place for such a description of achievements. Please state the purpose of your research here. This part can be transferred to the summary of the discussion.

Answer: Dear Review#4, the summary of work & contribution at the end of the Introduction section is a common writing manner. Though there may be more adequate writhing way, we believe this writing manner is acceptable for readers.

  1. Fig. 2 What is the unit?

Answer: Thank you. We explained in the corresponding sentences.

The normalized number of fires pixels was calculated as (N/A)×S, where N denotes the number of fire pixels in a latitudinal band, A denotes the area of land covered by this band (km2), and S is a scaling factor (1,000,000) to rescale the values to a more convenient range.

  1. Fig. 4 What process does this arrow mean? What is the relationship between the top and bottom images?

Answer: Thank you. We redraw Fig. 4(a) and added a sentence to explain the meaning of images in the Figure.

Top row is an active fire image, while bottom row the corresponding image before ignition.

  1. Fig. 10 What is the data source for the weather information?

Answer: Thank you. We added corresponding words in the caption of Fig. 10 and sentences to introduce the data source we used.

Round 2

Reviewer 3 Report

Dear Authors, 

Thank you for your answers and for work on the article. Most of my comments were taken into account, but the whole section remained unchanged (4.1.2 , 4.1.3). 

327 Please indicate which classes from the map (ESA landcover data) were merged to which when getting 5 final classes.

413 You have changed the data in Results but didn't the connected information in Discussion in 4.1.2.

Also, the name of Subarctic was inaccurately changed. Somewhere, it was not worth changing, but somewhere no changes were made. 

for instance 67  - Subarctic Circle?

91-96 - Arctic again

from previous revision 

452-453 Figure 11. It does not look like an cropland fire, but like a burned-out object, for example, a forested area or just a water body. Please check it. And if it is southern 60 latitude, it is not a good example.

 (c) Examples of fires which are affected by agricultural activities and hydrology (top row) – Need to be revised.

Author Response

Reviewer 3:

  1. 327 Please indicate which classes from the map (ESA landcover data) were merged to which when getting 5 final classes.

Answer: Thank you. We have added this information.

We reclassified land cover types in ESA Land Cover CCI Product [20], considering the classification quality of mosaic landscapes in high latitudes. Tundra and grassland were merged to tundra; scrubland and cropland were merged to scrubland/herbaceous; forest and wetland remained unchanged. The rest land cover types were merged to others.

  1. 413 You have changed the data in Results but didn't the connected information in Discussion in 4.1.2.

Also, the name of Subarctic was inaccurately changed. Somewhere, it was not worth changing, but somewhere no changes were made.

for instance 67  - Subarctic Circle?

91-96 - Arctic again

Answer: Thank you. We revised section 4.1.2 and 4.1.3 to keep consistency with the changes in section 3.4. We revised the full text to use Subarctic accurately.

  1. 452-453 Figure 11. It does not look like an cropland fire, but like a burned-out object, for example, a forested area or just a water body. Please check it. And if it is southern 60 latitude, it is not a good example.

(c) Examples of fires which are affected by agricultural activities and hydrology (top row) – Need to be revised.

Answer: Thank you. We checked the examples in original Figure 11c, and new selections of examples were made to redraw a new Figure 11. Corresponding text in section 4.1.2 and 4.1.3 were revised.